

# Honey bee inspired resource allocation scheme for IoT-driven smart healthcare applications in fog-cloud paradigm

Aasma Akram[1], Fatima Anjum[1], Sajid Latif[2], Muhammad Imran Zulfiqar[3] and Mohsin Nazir[4]

[1] Department of Computer Science, Lahore College for Women University, Lahore, Punjab, Pakistan
[2] Department of Computer Science, University of Arid Agriculture Rawalpindi, Rawalpindi, Punjab, Pakistan
[3] School of Computer Science and Engineering, Nanjing University of Science and Technology, Nanjing, JiangSu, China
[4] Department of Software Engineering, Lahore College for Women University, Lahore, Pakistan

## ABSTRACT

The Internet of Things (IoT) paradigm is a foundational and integral factor for the development of smart applications in different sectors. These applications are comprised over set of interconnected modules that exchange data and realize the distributed data flow (DDF) model. The execution of these modules on distant cloud data-center is prone to quality of service (QoS) degradation. This is where fog computing philosophy comes in to bridge this gap and bring the computation closer to the IoT devices. However, resource management in fog and optimal allocation of fog devices to application modules is critical for better resource utilization and achieve QoS. Significant challenge in this regard is to manage the fog network dynamically to determine cost effective placement of application modules on resources. In this study, we propose the optimal placement strategy for smart health-care application modules on fog resources. The objective of this strategy is to ensure optimal execution in terms of latency, bandwidth and earliest completion time as compared to few baseline techniques. A honey bee inspired strategy has been proposed for allocation and utilization of the resource for application module processing. In order to model the application and measure the effectiveness of our strategy, iFogSim Java-based simulation classes have been extended and conduct the experiments that demonstrate the satisfactory results.

# INTRODUCTION

Internet of Things (IoT) driven applications ingest data from devices connected to the internet and perform processing over it through the inter-collaboration of various modules (*Zulfiqar & Younis, 2024*). These modules compose the pipeline of data streams while executing certain logic to meet the business use case of a particular environment. In the medical field, automated monitoring process of patients locally or remotely

Corresponding author
Fatima Anjum,
fatima.anjum@lcwu.edu.pk

supports the doctors in providing better healthcare services and facilitates deciding critical conditions. IoT applications predominantly rely on fog-cloud computing infrastructure for computational and storage resources. The fog-cloud model offers a rich set of high-end resources and is considered the backbone for IoT applications, which get huge amounts of data from interconnected devices *e.g.*, sensors (*Taneja & Davy, 2017a*; *Kumar et al., 2024a*). CISCO revealed in the annual report that the number of IoT devices is increased to 30 billion by 2023 (*Walia, Kumar & Gill, 2023*). These huge numbers of devices generate enormous data, so large computational resources of cloud data centers are indispensable for providing long-term storage and computing. Fog computing functions as middleware between IoT and cloud resources to extend the computational capability closer to the IoT devices, so fog and edge computing can be used interchangeably in this context. Fog's distributed nodes fall in the local network of the system while cloud resources may exist in any remote location and can be accessed through the internet (*Latif et al., 2019*; *Latif et al., 2022*; *Latif et al., 2020*). The resource utilization model of the cloud is based on a pay-as-you-consume basis. On-demand cloud services facilitate scalable storage and processing services for IoT-driven applications (*Samriya et al., 2023*). To cope with the communication distance between the application and remote cloud setup, the fog concept comes in to handle latency-sensitive IoT applications (*Kumar et al., 2024b*; *Vadde & Kompalli, 2022*). Real-time data processing cannot be realized without fog middleware and QoS can be degraded due to the long distance of cloud setup. Fog-distributed nodes *e.g.*, routers, gateways, and switches can perform application modules placing and are supportive to mitigate the bandwidth and latency. Figure 1 depicts the fog-cloud architecture tier-wise and demonstrates the placing of various devices on tiers.

- *IoT tier:* This tier is comprised of data sources and sink devices *e.g.*, sensors and actuators. These instruments are built-in into devices, which are connected to the internet and distribute the data to the next level tier in the fog-cloud computing hierarchy.
- *Fog tier:* This tier extends the computing and processing capability of the cloud resources near IoT devices, so that gap could be bridged that happens due to the long distance between data generating sources and cloud sources. Fog resources have limited computational and storage capacity. Generally, the fog layer consists of gateways, routers, and access points (APs) devices. Proximity of edge devices with end-users is caused by a reduction in the latency and bandwidth *etc.*
- *Cloud tier:* High-end resources of cloud tiers provide the scalable service provisioning and long-term data storage. User applications consume resources of this tier on a pay-per-use model. Modules of IoT applications can be allocated to virtual resources according to customized data placement algorithms. In the fog-cloud computing stack, scheduling policies at each layer and the arrangement of resources have a significant impact on overall energy consumption and QoS.

Therefore, optimal management of computing resources on various layers for latency-sensitive IoT-applications processing is inevitable to get real-time statistics for the right decision-making. In this study, we propose the honey bee inspired resource allocation policy for distributed application modules. These modules get executed on fog devices until and

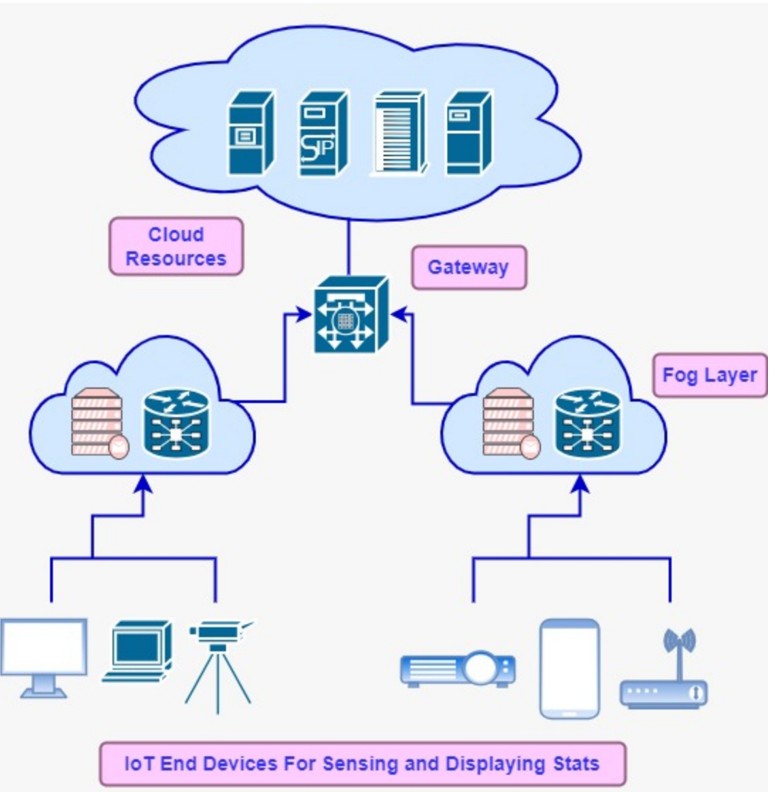

**Figure 1** **Fog-cloud model.**

unless the resource capacity of hosting goes beyond the requirement and subsequent modules may allocate cloud high-end resources. After getting data streams from the point of origin, modules perform certain processing and publish results to actuators for information. QoS for IoT-applications gets enhanced by allocating its distributed modules wisely to fog-cloud resources. Scheduling algorithms should be designed in such a way that the optimization could be ensured in terms of response time, bandwidth and compute utilization.

## Motivational factors and contributions

Motivational factors behind this research study are the combined utilization of diverse technologies *e.g.*, IoT, cloud and fog computing in optimal manner. These computing paradigms in modern applications are playing pivotal role particularly in the context of IoT driven applications. To harness the salient features of these technology in cost effective way for automation of smart applications in various sectors is challenging. The major reason is that different parts (AppModules) of the application need to deploy on various layers of this inter-collaborated setup in optimal manner and reliable flow of information amongst them require intelligent decision-making. So, this is where intelligent and optimal techniques come in. Some key motivational factors with respect to this study are mentioned below.

- Latency optimization: Real-time smart applications require low latency of information amongst modules so fog computing provide computation closer to the data sources devices as compared to cloud resources.
- Bandwidth constraints: Sending all information of IoT devices to cloud resources for processing consumes the high amount of bandwidth, by utilizing fog at edge filter the long retention information for cloud and remaining process at fog devices.
- Scalability: The distributed nature of fog computing provides better scalability as compared to its centralized cloud counterpart.
- Autonomy and reliability: As fog devices handle decision-making and data processing locally, which ensure the reliability even in case of bottleneck exist with cloud resources.

The following are main contributions of this article.

1. System model design and simulation-based implementation of IoT-driven healthcare applications by extending iFogSim simulation application programming interfaces (APIs).
2. Designed Honey bee inspired distributed application modules (AppModules) allocation technique to fog devices on the basis of objective function and problem formulation.
3. Presenting the comparison-based performance analysis with various baseline techniques on QoS parameters, which surpass these baseline algorithms.

The rest of the paper is organized as follows: Relevant work is discussed in 'Relevant work' while 'Proposed Architectural Design' presents the Proposed Architectural Design and related Algorithms. Simulation Experiments are discussed in 'Simulation Experiments' and 'Concluding remarks' present the Concluding Remarks.

## RELEVANT WORK

Designing and implementing smart applications in different fields *e.g.*, agriculture, medical and car parking, IoT and fog computing with cloud-centric resources are getting great attention in industry and academia. Various researchers have made tremendous contributions in this regard, which is highlighted as under.

In *Gupta et al. (2017)* and *Arshed & Ahmed (2021)*, the authors discussed the CloudSim-based simulation environment iFogSim to simulate the resource management policies for IoT applications. IFogSim is event driven libraries that utilize CloudSim as an event execution engine, which is developed in Java object-oriented programming language. The authors comprehensively discussed module mapping and placement strategies on edge and cloud resources. Researchers can extend the particular classes to implement their own customized algorithms to conduct various experiments in a controlled environment in repeatable manners. Huge upfront capital may be consumed while establishing a real test-bed so simulation is the most desirable choice for researchers. Different entities of IoT applications and cloud-fog resources can be modeled hassle-free. Cloud and Edgeward module placement techniques are also discussed to demonstrate the QoS impact along with some case studies.

Dynamic resource allocation to application modules and rivals has been discussed in *Gao et al. (2019)*. Traffic prediction with this allocation scheme is defined as the stochastic

problem of network optimization. Simulation results demonstrate the improvement in the consumption of power during stable processing delay.

*Li & Huang (2017)* have introduced the Markov decision process (MDP) based scheduling and resource management policies for IoT applications to reduce the energy consumption and delay of task execution delay.

Dynamic policy for allocation of resources has been proposed in *Ni et al. (2017)*. Petri nets (PTPNs) and the completion of jobs by fog nodes are the major factors of this policy. The job scheduling algorithm has been present for latency-sensitive applications in *Gupta et al. (2017)*, which demonstrates the application module (AppMondule) placement on the edge devices effectively rather than placement on cloud devices. EEG tractor game is simulated to compute various metrics *e.g.*, bandwidth usage, consumption of energy, and loop delay using the FCFS algorithm on the fog-cloud layer.

*Bittencourt et al. (2017)* used different strategies *e.g.*, FCFS, Concurrent, and delay-priority for module placement against applications like latency-sensitive and latency-tolerant in the fog paradigm. Each scheduling strategy has its problems while allocating application modules. When the ratio of scheduling modules creases in concurrent strategy, it indicates a resource contention issue which is overcome by using FCFS through offloading modules from fog devices to cloud resources. However, the remote location of the cloud setup causes delays and increased bandwidth usage from latency-sensitive applications. To resolve the issue, a delay priority policy has taken place to dispatch modules based on delay sensitivity. Despite this, in the case of increasing modules of applications, this approach does not meet the required QoS factors and causes the increment in overheads *e.g.*, delay, cost, and high bandwidth usage of cloud resources.

A prioritized job scheduling technique is proposed by *Choudhari, Moh & Moh (2018)* to minimize the overall response time and cost. Priorities of jobs are calculated by the defined deadlines of all jobs. These priorities play a critical role in assigning jobs in the fog layer. This layer contains multiple fog nodes with heterogeneous computational and storage resources. If these resources get saturated, jobs are allocated to the upper cloud layer. The primary goal of this approach is to balance the allocation of memory and execution time and other metrics *e.g.*, utilization of network and consumption are not taken into account.

The authors investigated Infrastructure as a service (IaaS) provisioning at the commercial level in *Durillo & Prodan (2014)*. To make the trade-off amongst appropriate solutions, the Pareto front method is used as the decision-making tool. By using this method, the cost of scheduling has been halved; however, a 5 percent increase in make-span was observed. The knapsack scheduling algorithm has been proposed in *Lao, Zhang & Guo (2012)* to achieve the minimum completion time (MCT) objective for transferring parallel video in the cloud. To map the tasks to the number of segments on powerful systems, the max–min algorithm is executed and then scheduling of those segments is performed by MCT. Experiments depicted that max–min outperformed the MCT in both cases.

The algorithm for placement of IoT application modules is proposed in *Nashaat, Ahmed & Rizk (2020)* to enhance the system performance. However, in case of a slight increment in power consumption, it's required to check its impact. In this technique, application modules are placed on devices, which are suitable in terms of proximity, computational

resources, and response time. Modules get prioritized to be placed concerning criteria *e.g.*, application usage, QoS violation, and user expectations. Research in large-scale IoT and context-aware applications with fog computational resources allocation to modules of this application is still at the initial stage (*Mach & Becvar, 2017*; *Brogi & Forti, 2017*; *Liu et al., 2019*).

*Azizi, Khosroabadi & Shojafar (2019)* proposed a QoS-aware modules placement algorithm on edge devices for delay-sensitive applications to minimize the response time and cost in the fog-cloud paradigm. Similarly, a module placement strategy has been proposed in *Mahmud, Ramamohanarao & Buyya (2018)* along with the optimization of resources without suffering from QoS parameters. There are two algorithms used in this system configuration to achieve QoS and module placement to cater to the optimal response time for application requirements. Contextual information-based distributed application strategy proposed in *Tran et al. (2019)* for fog paradigm. The algorithm utilizes the contextual information to assign modules on fog resources. This information includes the QoS deadline, proximity of resources, and service type to provision the resources for IoT-driven application modules. Experiments in this approach depict the reduced response and optimal efficiency of the fog network. However, by increasing nodes in the fog network in this setup, calculation time exponentially increases in this solution.

*Taneja & Davy (2017b)*, proposed an application module placement policy for fog-cloud resources based on the available capacity of network nodes. This strategy utilizes the network resources effectively. The proposed algorithm functions for placing application services as it iterates over entire available nodes to find the most eligible one unless it is exhausted. To consume the network resources effectively in a way that application delay could be minimized, a heuristic approach for heterogeneous resources has been proposed in *Xavier et al. (2020)*.

*Gupta et al. (2021)* and *Atlam, Walters & Wills (2018)* shed light on fog computing paradigm characteristics and applications that use fog resources with IoT data flow amongst components of these applications. There is much literature available, which shows that direct interaction with cloud-based resources from IoT-driven applications is much higher as compared to resources of fog layer. Due to the proximity of IoT devices, the fog layer minimizes the bandwidth and supports the scalability factor. In the fog computing paradigm, utilization of the network is a vital metric for real-time time-critical applications and it must be minimized.

To maintain the security protocols of citizens and to meet privacy requirements, geo-distribution of fog resources is used by researchers. Fog based monitoring system in the proposed setup minimizes the general conflict of the public. An artificial intelligence (AI) based fog system has been proposed by *Munir et al. (2021)*. This study was conducted to provide feedback to emergencies rapidly with awareness of the situation state using urban surveillance through the air medium. Thirty-seven percent latency improvement is shown over the cloud paradigm.

Similarly in the education sector, a cyber-physical system (CPS) is proposed for monitoring purposes in the educational sector intelligently in the article (*Singh & Sood, 2020*). The objective of this study was to reduce the consumption of energy and delay.

The Optical Fog node as middleware is deployed between cloud setup and data generating sources. A comparison between fog computing and the cloud is also discovered in another study by *Mondal et al. (2022)*. Vehicle detection-based model for smart city planning (*Chen et al., 2017*).

*Walia, Kumar & Gill (2023)* presented resource management issues and challenges regarding fog/edge comprehensively. They have categorized issues into various domains *e.g.*, scheduling of resources, placement of services and load-balancing *etc*. Authors emphasized on machine learning (ML) and traditional approach to solve issues related to different domains and discussed the state-of-the-art literature review with different QoS parameters. Research directions towards latest technologies like software defined network (SDN), 5G and serverless computing has also been discussed. Research students can get directions from this detailed study in fog computing.

This research study (*Kumar et al., 2024a*) present the optimization of QoS by discussing MEC architecture that depict the bi-objective optimization problem that include also cost minimizing, energy and deadline constraints. GA-PSO meta-heuristic optimization technique is embedded in MEC architecture. In *Samriya et al. (2023)*, adversarial machine learning (AML) has been proposed to implement the IoT devices security from modern threats. This study focuses to utilize AML techniques in smart applications for health-care domain.

*Kumar et al. (2024b)* proposed autonomic edge-assisted cloud-IoT framework for smart application in health-care sector, which uses Random Forest and logistic regression grid (RF-LGR) technique at edge network for heart disease analysis and for the improvement of various parameters and compared with K-nearest network (KNN), LR and Random Forest (RF) algorithms.

In above presented literature, we have explored those researchers has not used nature-inspired technique except few for distributed fog setup to achieve optimization in terms of compute and data transferring rate. Most of the works presented classical algorithms for allocation of resources, which are not as scalable as our proposed methodology. The summary of traditional techniques is shown in Table 1. Our goal in this study is to target the distributed fog devices where least mean utilization is observed dynamically with two parameters *e.g.*, compute and bandwidth. Application modules in proposed scheme are light weighted so we did not take memory as parameter in mean utilization. Applications like traffic signaling and health-care services demand the prompt response amongst inter-modules communication so delay can be critical for decision-making. Hence, proposed allocation technique is more optimal than classical algorithms. The proposed design and algorithms are discussed in next sections.

## PROPOSED ARCHITECTURAL DESIGN

The combination of both fog-cloud paradigms empowers IoT-driven applications by enabling low latency due to the close vicinity between data sources (IoT sensors) and processing modules on fog devices. Cloud resources in this schema provide centralized storage services for distributed end data source devices. This distributed architecture

**Table 1 Strength & limitations of traditional techniques.**

| Sr. no | Reference & Year | Parameter | Strength | Limitations |
|---|---|---|---|---|
| 1 | 2019 (*Atlam, Walters & Wills, 2018*) | Stochastic problem | Overcome network issue | Static scheduling |
| 2 | 2017 (*Azizi, Khosroabadi & Shojafar, 2019*) | Energy | Reduce consumption | Less scalability |
| 3 | 2019 (*Bittencourt et al., 2017*) | Latency | Dynamic nature | Less throughput |
| 4 | 2019 (*Gao et al., 2019*) | Delay factor | Robust mapping heuristic | Slow convergence |
| 5 | 2020 (*Gupta et al., 2021*) | Completion time | Minimize completion time | Less optimal mapping |

comprises different tiers with different natures of computing devices, which compose the fog-based IoT ecosystem. Apart from physical computing and storage devices, logical components *e.g.*, application modules and application edges are also in-text ingredients of this distributed schema. Amongst all devices of different tiers, gateway devices interconnect the different tiers for the exchange of data and enable the conversion of protocols across network segments. The node capacity in the proposed schema is defined through attributes *e.g.*, CPU, RAM, and Bandwidth. It is pertinent to mention that the algorithm that will map the resources should be able to scale additional attributes *e.g.*, storage if required by the application module for long retention over the cloud.

Figure 2 depicts the proposed design of implementation with various layers. The capacity of the any computational fog device $F\_i$ in network can be defined with different attributes as $Cap(F\_i) = < CPU\_i, RAM\_i, BW\_i >$ and the total computing capacity of the fog network is represented by sum of the capacity of all devices. Application modules harness the available capacity of resources on tiers. The deployed application is based on the Distributed Data Flow Model (DDF) in this model. As components of the application are distributed modules interaction amongst them through DDF produces better results. These modules are modeled as directed acyclic graph (DAG) in which vertices represent the different processing modules and edges represent the flow or data dependency. These modules process ingested data and produce output as input data for another module in DAG. As application DAG A consists of vertices and edges so can be defined mathematically as $A = < V, E >$. Each component (module) requires resources in terms of computing (CPU), memory (RAM), and bandwidth (BW) for execution and data transfer. The algorithm for placement of these modules is evenly scalable even when adding more attributes to meet the requirements of the application. If vertica $V\_i$ denote *appModule\_i* the resource requirement of this particular module can be defined as $Resource\_Req(Vi) = < Compute\_CPUi, Memory\_RAMi, Bandwidth\_BWi >$. Figure 3 represents the flow of information amongst different modules of application.

In our proposed model various modules inter-collaborate with each other and perform different functions for processing the data collected from devices. The data flow of Application modules in the proposed configuration is unidirectional. Each component (module) receives the data stream from its previous component performs certain operations and dispatches the result to the next connected module in a directed graph. The following modules compose the IoT-driven healthcare application in our model, which utilizes the

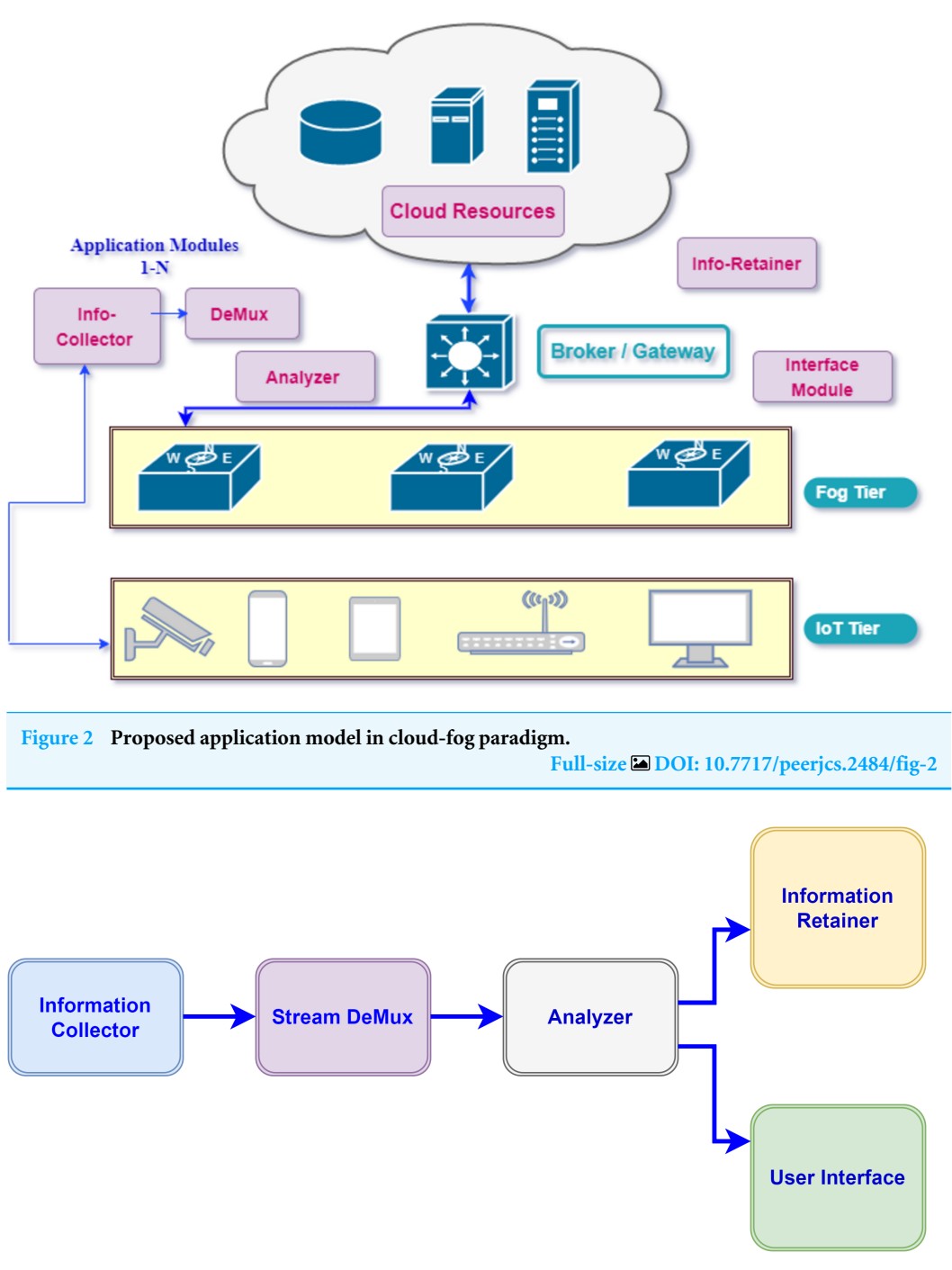

**Figure 2** Proposed application model in cloud-fog paradigm.

**Figure 3** Information flow amongst modules.

fog-cloud paradigms for storage and computational requirements. Fog resources extend the capabilities of the cloud close to the data sources to bridge the long distance of clouds.

- Information collector: This module of the application receives the data stream from IoT devices *e.g.*, sensors and applies the data compression standard before dispatching

to the destination module in a directed acyclic graph for processing. The information collector module also maintains the buffer for received data for a certain time to avoid the packets discarding from the queue.

- Stream de-multiplexer: De-multiplexing of the received stream is performed by this component to segregate the various statistics *e.g.*, pulse rate and heart-beat rate of patients and forward to next for automated analysis.
- Analyzer: Some sub-modules perform functions in the analyzer module to make the analysis of different sorts of collected data through the automation process and send the result-end to the next connected modules.
- Information retainer: To retain the information for a long time and keep the historical data of patients, this module performs the tasks and sends the information to cloud storage.
- User interface: To perform the monitoring by medical staff this module plays an important function in displaying the data on devices *e.g.*, actuators.
- Broker: The brokering component performs the inter-communication between Cloud and Fog tiers through the gateway device.

## Communication mode

All modules of application inter-collaborate with each other through exchange of messages (events) and implement the transport layer security (TLS) protocol at transport layer. This protocol ensures the secure communication amongst application modules (AppModules) through encryption from end to end. Moreover, required resources for the processing of these modules in fog network and cloud site are non-preemptive, which never interrupt the processing till completion.

## Modules allocation strategy

Allocation methodology of application modules assignment is inspired from the foraging behavior of honey bees, which mimics the food searching patterns of bees in Algorithm 1. Scout bees search randomly and forager bees harness the best source of food as our aim is also target that device, which has less mean utilization (compute and bandwidth). So that earliest response time and completion time could be observed. Pro and cons of this technique is discussed below. Algorithm 2 is used as procedure of calculating the utilization of fog nodes. The algorithm iterates over the available fog nodes and calculates the utilization of load and bandwidth then computes the least mean value of both to place the application module. This process continues until all modules of the application get placed over nodes.

## Pros

- Decentralized nature: Algorithm function in a decentralized manner as module assign across multiple devices without any centralized controller entity. This factor causes of scalability and fault-tolerance.
- Efficient utilization of resources: Allocation of resources is performed on the basis of fitness level *e.g.*, earliest completion time, minimum utilization and energy consumption, which leads to optimal level of utilization of resources.

**Algorithm 1** Application Modules Mapping

1: **function** *ModuleMapping*(*appModulesList*, *fDeviceList*)
2:     *numScouts* ← 15              ▷ Number of Scout Bees
3:     *maxIterations* ← 70
4:     *numForagers* ← 50              ▷ Forager Bees
5:     *Random rand* ← *createObject*(*Random.Class*)     ▷ Random Class Object
6:     **for** Itr = 1 to *maxIterations* **do**
7:         **for** Sct = 1 to *numScout* **do**
8:             **for** *AppModule* ∈ *appModuleList* **do**
9:                 *FogDevice fd* ← *fDeviceList.get*(*rand.nextInt*(*fDeviceList.size*))
10:                 *AppModule.setFogDeviceId*(*fd.getId*())
11:             **end for**
12:         **end for**
13:     *bestFitness* ← *calculateFitness*(*appModuleList*, *fDeviceList*) ▷ Calculating fitness of resources
14:         **for** frgBee = 1 to *numForager* **do**     ▷ Foragers Bees Harness the best solutions
15:             **for** *AppModule* ∈ *appModuleList* **do**
16:                 **if** *rand.nextDouble*() < 0.1 **then**
17:                     *FogDevice fd* ← *fDeviceList.get*(*rand.nextInt*(*fDeviceList.size*))
18:                     *AppModule.setFogDeviceId*(*fd.getId*())
19:                 **end if**
20:             **end for**
21:         *newFitness* ← *calculateFitness*(*appModuleList*, *fDeviceList*)
22:         **if** *newFitness* < *bestFitness* **then**
23: *bestFitness* ← *newFitness*
24:         **else**             ▷ Revert changes if fitness is worse
25:             **for** *AppModule* ∈ *appModuleList* **do**
26:                 *FogDevice fd* ← *fDeviceList.get*(*rand.nextInt*(*fDeviceList.size*))
27:                 *AppModule.setFogDeviceId*(*fd.getId*())
28:             **end for**
29:         **end if**
30:         **end for**
31:     **end for**
32: **end function**

- Scalability factor: The technique enhances scalability factor easily with increasing numbers of tasks and resources and is suitable for distributed environment.
- Adaptive: It constantly evaluates the dynamism of the environment and adapts to changes in load at runtime before assigning modules.
- Self organizing behavior: It manages resources autonomously without external intervention as it mimic self-organizing behavior of bees.

---

**Algorithm 2** Calculating Fitness of Resources

---

1: **function** *calculateFitness(appModulesList, FDeviceList)* ▷ Find the fitness of Resources
2:     *totalMeanUtilization* ← 0
3:     **for** *FDevice* ∈ *FDeviceList* **do**
4:         *meanUtilization* ← 0
5:         **for** *AppModule* ∈ *appModuleList* **do**
6:             **if** *AppModule.getFdID() = FDevice.getID()* **then**
7:
8:                 *CpuUtilization* ← *FDevice.getTotalUtilizationOfCpuMips(AppModule)*
9:                 *BWUtilization* ← *FDevice.getTotalUtilizationOfBW(AppModule)*
10:                 *meanUtilization* ← *meanUtilization* + (*CpuUtilization* + *BWUtilization*)/2
11:             **end if**
12:         **end for**
13:         *totalMeanUtilization* ← *Math.min(totalMeanUtilization, meanUtilization)*
14:     **end for**
15:     *return totalMeanUtilization*
16: **end function**

---

## Cons

- Communication overhead: Bees communicate the information of distributed amongst resources frequently, which creates the overhead in large scale fog devices network.
- Complexity: The honey bee inspired algorithm creates complexity due to its bees like behaviors of searching food. In order to implement this behavior multiple agents (bees) have to manage and initialize in the system.
- Dependency: Certain parameters are inevitable in these types of techniques like number of forager agents, scout bees *etc.* and setting of these parameters can be challenging.
- Slow convergence: In order to find optimal solution in some cases algorithm may take long time while competing for resources by many tasks.

## Mathematical model

The least mean utilization in the proposed technique also helps to balance the load on devices of the fog network, which reduces the makespan and response time. The completion time of $AppModule_i$ on device $FD_j$ as $CT_{ij}$, so Makespan is the overall module completion time and can be calculated through the following Eq. (1).

$$makespan = max\{CT\_ij \mid i \in AppModule, i = 1,2,....n \text{ and } j \in FD j = 1,2,....m\} \quad (1)$$

and response time is the amount of time taken in application module submission and the first response is obtained. Let $AppModules = \{AppModule_1, AppModule_2....AppModule_n\}$ and these modules will be executed on set of fog devices $FD = \{FD_1, FD_2....FD_n\}$. All AppModules processing on these devices is based on a non-preemptive pattern, which means that the execution of the module will not be interrupted. The processing time of AppModule in $FD_j$ can be denoted as $P_{ij}$. Processing time of $AppModule_i$ on $FD_j$ is denoted

as $P_{ij}$ and processing time of all *AppModules* in $FD_j$ can be defined by Eq. (2).

$$P_j = \sum_{i=1}^{n} P_{ij} \, j = 1, \ldots, m. \tag{2}$$

Equation (3) is obtained by reducing the value of $CT_{max}$ and Eq. (4) is implied through Eq. (2) and Eq. (3).

$$\sum_{i=1}^{n} P_{ij} CT_{max} \leq j = 1, \ldots, m \tag{3}$$

$$\Rightarrow P_{ij} \leq CT_{max} \, j = 1, \ldots, m. \tag{4}$$

The processing time of application modules (AppModules) reflects the variation from one fog device to another based on capacity. The load of fog device $FD-j$ can be calculated based on executing millions of instructions of application module (*AppModule*) and the capacity of fog device is calculated through Eq. (5).

$$FD_j = PE_{numj} x PE_{mipsj} + FD_{bwj}. \tag{5}$$

$PE_{numj}$ denotes quantity of processors and $PE_{mipsj}$ has been used to indicate the processing power of each processing element (PE) and $FD_{bwj}$ is the bandwidth capacity of communication channel amongst other fog devices in the network. The capacity of all fog devices participating in the network can be also, calculate to know the total utilization of computational resources.

$$Total_{cap} = \sum_{i=1}^{m} FD_j. \tag{6}$$

The total computational capacity of fog network can also be derived through Eq. (6) and measurement of load on fog device $FD_j$ is calculated through Eq. (7). Application modules (AppModules) length in terms of Millions Instruction (MIs) allocated to fog device $FD_j$ in any given time is called workload.

$$Load_{FDi,t} = N(AppModule_{MI}, t)/S(FD_j, t). \tag{7}$$

Equation (7) states the execution of the application module's millions of instruction (MI) by service rate $S$ of fog device $FD_j$ in time $t$ is called load on single fog device. The total load of all devices is derived by Eq. (8).

$$Total_{Load} = \sum_{i=1}^{m} Load_{FDi}. \tag{8}$$

Similarly, processing time of fog device $FD$ is computed through Eq. (9).

$$PT_i = Load_{FDi}/C_i. \tag{9}$$

The overall processing time of all fog devices $FDs$ is $PT = L/C$ and utilization of bandwidth by application module on fog device is calculated as in Eq. (10)

$$BWUtil_{FDi,t} = EventBytes_{AppModulei}/UPLinkCapacity_{FDj}. \tag{10}$$

Event bytes indicate the length of message, which particular application module will send to next module as output on network link at given time $t$. Mean utilization of fog device $FD_j$ resource will be obtained through Eq. (11).

$$Mean_{Utilizatiion} = Load_{FDi,t} + BWUtil_{FDj,t}/2. \tag{11}$$

## Simulation experiments

To conduct the experiments of the proposed design, the application programming interface (API)/Libraries of iFogSim have been harnessed, which are object-oriented and are written in Java programming language. Due to their object-oriented nature, these libraries are easily extensible. Researchers can extend these classes according to their desired parameters and algorithms. Simulation experiments can be performed in a repeatable manner in a controlled environment and are consider cost-effective as compared to real test beds. Mainstream classes, which researchers extend to model their proposed entities are the following. iFogSim utilizes the CloudSim simulation engine for inter-entities communication and events passing.

- SimEntity: This class is the mainstream class of CloudSim and several classes of iFogSim implement this one to send the various types of events to each other.
- FogDevice: This class is use to model the computing resources of fog network and perform the computation of application modules.
- Sensor: To simulate the hardware devices cameras and sensors, researchers use this class that originates the data in the form of tuples.
- Actuator: This class is used for any hardware that displays statistics after processing tuples received from sensors.
- ModuleMapping: It is an abstract class used for providing customized implementation of mapping of modules on fog nodes.
- AppModule: IoT application comprised of various modules, AppModule class is used to model the different modules of application.
- FogBroker: Basically, this class simulates the brokering resources of the fog network and perform the computation of application modules.

Figure 4 depicts the network topology of cloud and fog nodes in our model and both networks are connected through a proxy server/brokering component.

Figure 5 indicates the processing time with various numbers of applications through applying different application module mapping strategies. Since the proposed methodology calculates the least mean utilization of device resources and allocates the module to the least utilized resource processing time remains low comparatively to other mapping techniques. Figure 6 depicts the bandwidth utilization of devices, which application modules (AppModules) utilize while exchanging information in a fog network. Since these modules function as distributed data flow (DDF) so output of some modules is the mandatory input of other modules and persists in the inter-dependency. Figure 7 illustrates the latency variations, representing application response times for varying numbers of applications in our proposed design, compared to both the mapping (Default)

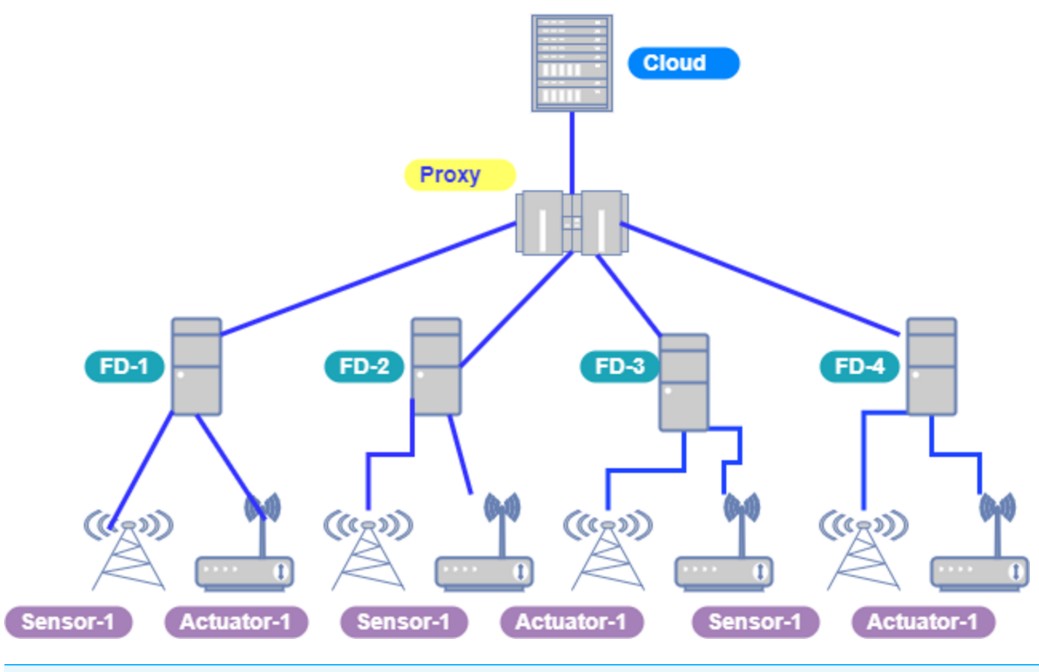

**Figure 4** Cloud-fog topology setup.

and cloud-only designs. Our design outperformed both baseline designs. In our simulation setup, hierarchical topology has been configured, which is depicted in Fig. 4. In this topology configuration root indicates the cloud setup and leaves are considered edge devices like gateway and these devices are configured with different compute, bandwidth, and storage capacities. Various modules of IoT applications get executed on these devices and consume their resources. We have configured simulation parameters for the cloud data center, fog devices are shown in Table 2 and Table 3, respectively. These parameters can vary as desired requirements. The distance of a cloud data center is long compared to fog devices from data-producing sources. Fog computational devices are not as high-end as cloud resources and can keep the retention of data for a long time.

Table 4 shows gateway device parameters, and used for inter-communication between cloud and fog network. Gateway devices are configured and deployed at second level in proposed model.

Devices in the model interact through the communication of application modules deployed on them. These modules generate workload in the form of tuples (Millions of Instructions) and traverse these instructions with each other to perform certain task executions. In the upward direction (cloud & fog) these instructions are called Tuples-Up and Tuples-Down are those instructions, that traverse amongst application modules.

## CONCLUDING REMARKS

In this study, we propose the honey bee inspired modules placement strategy for health-care latency-sensitive IoT applications. This application requires an optimal response resource utilization, so that, efficient data flow amongst distributed modules could be ensured and

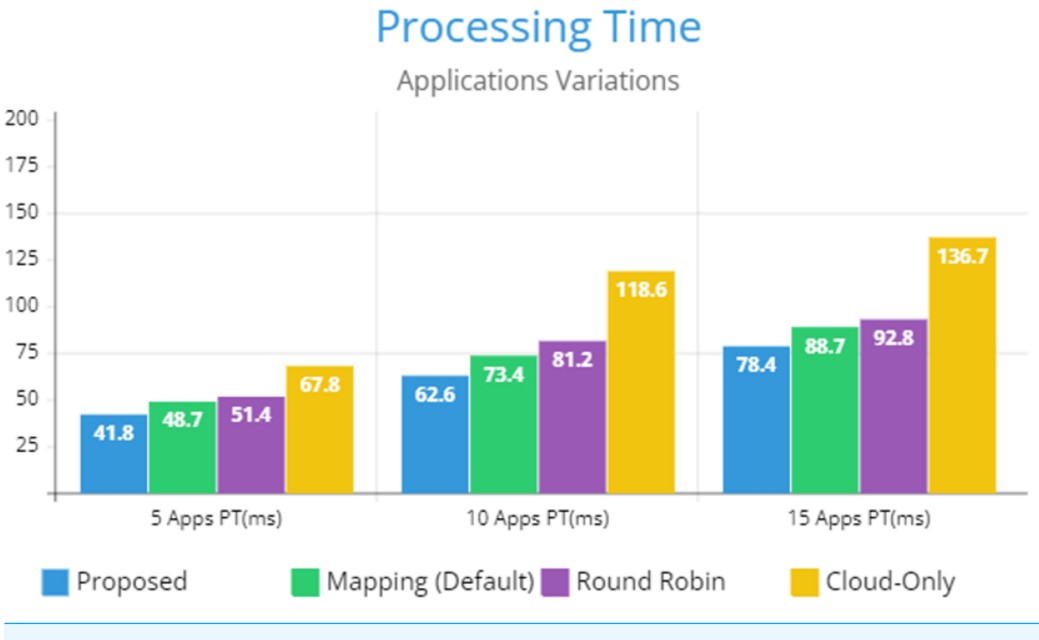

**Figure 5** Applications processing time.

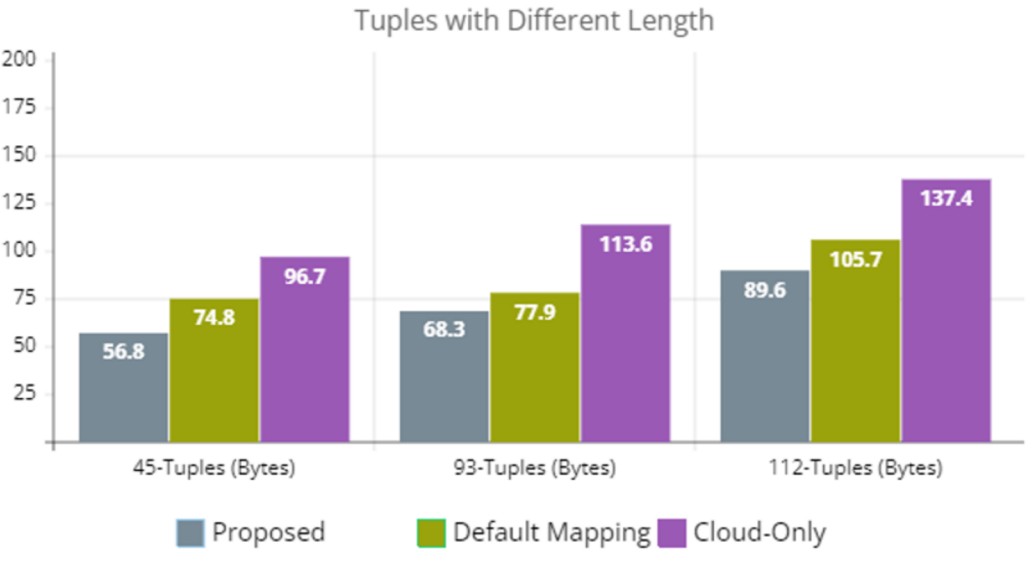

**Figure 6** Bandwidth consumption.

exact decisions could be taken because of the patient's criticality. The proposed algorithm places the module on fog computational resources based on an existing load of resources and the capacity of the earliest execution time. To evaluate this algorithm, resources with different capacities were taken in a simulation environment and performed the tests along with other placement techniques. We have performed various tests with a varied number
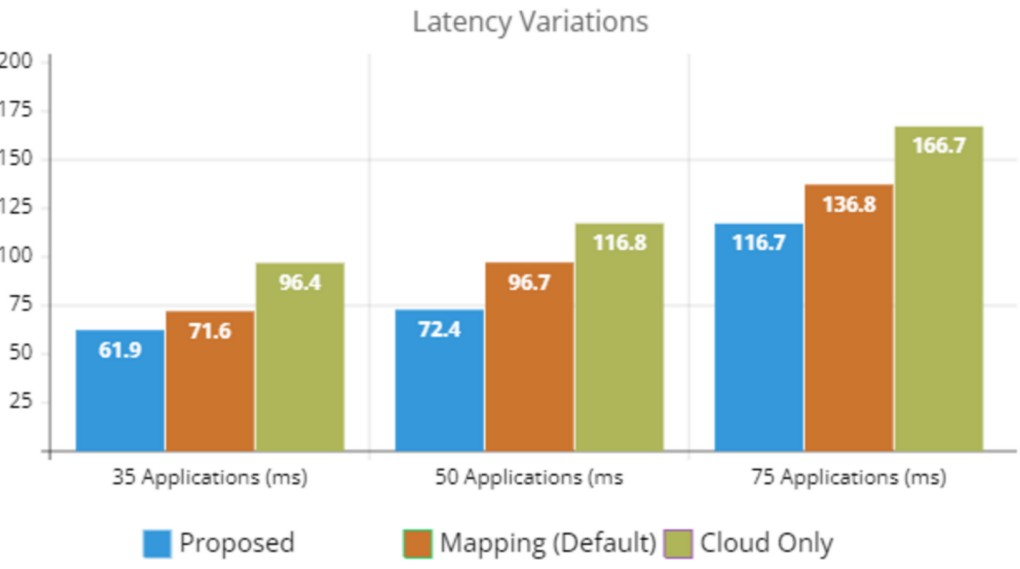

**Figure 7** Latency variations.

**Table 2** Cloud data center parameters.

| Parameters | Value |
| --- | --- |
| Name of device | Cloud |
| Millions instruction per second | 56,320 |
| RAM | 60 GB |
| Uploading bandwidth | 110 Mbits/sec |
| Downloading bandwidth | 12 Gbits/sec |
| Parent level | 0 (Cloud is Top) |
| Rate per processing usage | 0.02 |
| Busy power | 1,532 W |
| Idle power | 1,225 W |

of applications. The proposed strategy has been examined against other techniques *e.g.*, round-robin, mapping algorithm, and cloud-only placement. Simulated results show the improvement in latency, processing time, and consumption of bandwidth by placing modules through a proposed algorithm. This algorithm maps the module after computing the mean utilization of participating devices in the fog network and targets the device, which has the least mean utilization on the principle of honey-bee behavior. The testing scenario of this study to examine results has been simulated in an iFogSim simulation environment that produced satisfactory results. In the future, we intend to apply the proposed approach to other smart applications, including car parking systems and traffic signaling systems, to assess its overall impact and implications.

**Table 3  Fog devices parameters.**

| Parameters | Value |
|---|---|
| Name of device | Broker |
| MI per second | 3,100 mips |
| RAM | 6 GB |
| Uploading bandwidth | 11 Gbits/sec |
| Downloading bandwidth | 11 Gbits/sec |
| Parent level | 1 (Cloud) |
| Rate per processing usage | 0.02 |
| Busy power | 107 W |
| Idle power | 75 W |
| Latency (Cloud & Broker) | 110 ms |

**Table 4  Gateway devices parameters.**

| Parameters | Value |
|---|---|
| Name of device | Broker |
| MI per second | 3,100 mips |
| RAM | 6 GB |
| Uploading bandwidth | 11 Gbits/sec |
| Downloading bandwidth | 11 Gbits/sec |
| Parent level | 1 (Cloud) |
| Rate per processing usage | 0.02 |
| Busy power | 107 W |
| Idle power | 75 W |
| Latency (Cloud & Broker) | 110 ms |

### Funding

The authors received no funding for this work.

### Competing Interests

The authors declare there are no competing interests.

### Author Contributions

- Aasma Akram conceived and designed the experiments, performed the experiments, analyzed the data, performed the computation work, prepared figures and/or tables, and approved the final draft.
- Fatima Anjum performed the experiments, prepared figures and/or tables, authored or reviewed drafts of the article, and approved the final draft.

- Sajid Latif conceived and designed the experiments, performed the experiments, performed the computation work, prepared figures and/or tables, and approved the final draft.
- Muhammad Imran Zulfiqar analyzed the data, authored or reviewed drafts of the article, and approved the final draft.
- Mohsin Nazir analyzed the data, authored or reviewed drafts of the article, and approved the final draft.

## Data Availability

The code is available in the Supplemental File.

## Supplemental Information

Supplemental information for this article can be found online at http://dx.doi.org/10.7717/peerj-cs.2484#supplemental-information.

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
