# Peer review of "Honey bee inspired resource allocation scheme for IoT-driven smart healthcare applications in fog-cloud paradigm"

_PeerJ Computer Science, doi:10.7717/peerj-cs.2484_

## Round 0.1 · original submission · Major Revisions

· Academic Editor

Major Revisions

Dear authors,

Thank you for submitting your article. Feedback from the reviewers is now available. It is not recommended that your article be published in its current format. However, we strongly recommend that you address the issues raised by the reviewers, especially those related to readability, experimental design and validity, and resubmit your paper after making the necessary changes. Before submitting the paper following should also be addressed:

1. What is your central research question? How does your subject relate to your research problem? What methods should you use to analyse your research problem? Why is your research important and why should someone reading the proposal care?

2. How the honey bee-inspired strategy is adapted to the focused problem is not clear. "Honey bee" only exists in the Abstract section of the paper.
3. Pros and cons of the methods should be clarified. What are the limitation(s) methodology(ies) adopted in this work? Please indicate practical advantages, and discuss research limitations.

4. Explanation of the equations should be checked. All variables should be written in italics as in the equations. Their definitions and boundaries should be defined. Necessary references should also be given.

5. Reviewer 1 has asked you to provide specific references. You are welcome to add them if you think they are relevant. However, you are not obliged to include these citations, and if you do not, it will not affect my decision.

Best wishes,

·

Basic reporting

The authors objective is to deploy the application processing modules on Fog resources for optimal execution in terms of latency, bandwidth, and earliest completion time. The honey bee-inspired strategy has been proposed for allocation and utilization of the resource for application module processing. My comments are given below:
1. The authors should carefully proofread this paper and correct all the typos in the revision.
2. Authors need to add the motivation for the article in introduction section.
3. Authors need to rewrite the abstract to focus more contributions.
4. The need of fog computing is not described properly in introduction, authors discuss with the help of real time example.
5. Authors need to add state of art comparison table along with its advantages and disadvantages in section 2.
6. Problem formulation is cloud computing based, not integrated cloud fog. Author go through some reputed journals and improve the problem formulation.
7. Authors need to discuss proposed framework along with its components, how offloading and resource allocation decision has made.
8. Many researchers has applied GA and other metaheuristic algorithm, how your work different from others.
9. There are lots of techniques have been proposed by various authors in recent years, how your work is novel, could you justify.
10. Check the symbols and equations used in problem formulation.
11. The experimental results are not looking promising, authors discuss with more details.
12. Authors can add the limitations of proposed techniques.
13. Literature needs to be improved by incorporating some quality papers like
(i) AI-empowered Fog/Edge Resource Management for IoT Applications: A Comprehensive Review, Research Challenges and Future Perspectives
(ii) Deadline-aware Cost and Energy Efficient Offloading in Mobile Edge Computing
(iii) Adversarial ML-Based Secured Cloud Architecture for Consumer Internet of Things of Smart Healthcare
(iv) Autonomic Edge Cloud Assisted Framework for Heart Disease Prediction using RF-LRG Algorithm

Experimental design

mentioned in basic report

Validity of the findings

Please find the comments in basic reporting

Additional comments

As above

·

Basic reporting

1.Authors own contribution is missing.
2. Authors have mentioned the healthcare application in tittle, but not mentioned how this proposed approach is helpful in smarthealthcare system.
3. lots of grammatical error.
4. Latest references is missing in work.

Experimental design

1. own menthodology is missing in the proposed approach.
2. Algorithm has not provided any new research or any innovation.
3. There is requirement of Background chapter, who efficiently explain the Honey Bee algorithm.
4. in tittle author have mentioned Honey Bee algorithm, but where its used not clear in manuscript.

Validity of the findings

1. written abstract and conclusion is weak.
2. used dataset is not mentioned.

Additional comments

1. overall written manuscript is not linked properly.
2. paper is rejected.

Reviewer 3 ·

Basic reporting

1. Abstract must define the context, the problem discussion, solution proposed, results, findings and implications of research.
2. A more detailed and refined comparison is required among the proposed approach and existing approaches.

Experimental design

Scalability issue is not addressed anywhere in the paper. How the proposed approach will work in different healthcare scenarios if it get employed in real life scenarios.

Validity of the findings

One more parameter of evaluation must be added that validated the appropriate resource allocation is done or not using the proposed approach.

Additional comments

Results must highlight the percentage of improvement the proposed approach made over existing ones.

---

## Round 0.2 · accepted · Accept

· Academic Editor

Accept

Dear authors,

Thank you for the revision. Two of the original reviewers did not respond to the invitation to review the revised paper. Another reviewer thinks that the paper is acceptable for publication. I also think that the paper is sufficiently improved and seems acceptable for publication.

Best wishes,

·

Basic reporting

Authors have addressed all the comments successfully and paper may be accepted for article.

Experimental design

Good

Validity of the findings

results are validate at simulation environment

Additional comments

Authors have addressed all the comments successfully and paper may be accepted for article.